# Optimization of Iron Recovery from BOF Slag by Oxidation and Magnetic Separation

## Mo Lan, Zhanwei He and Xiaojun Hu *

State Key Laboratory of Advanced Metallurgy, University of Science and Technology Beijing,
Beijing 100083, China; g20199312@xs.ustb.edu.cn (M.L.); hezhanwei@ustb.edu.cn (Z.H.)
* Correspondence: huxiaojun@ustb.edu.cn

**Abstract:** In order to solve the problem of solid waste pollution of basic oxygen furnace (BOF) slag in the metallurgical process, this paper took BOF slag as the research object, and carried out oxidation reconstruction of BOF slag and alcohol wet magnetic separation recovery of iron phase, so as to efficiently recover and utilize BOF slag. In the early stages, the research group realized the transformation from weak magnetic iron oxide to strong magnetic magnesia-iron spinel phase in BOF slag through oxidation reconstruction experiments under different technological parameters. On this basis, different conditions in the magnetic separation process were adjusted to achieve the optimal iron recovery rate and grade in this paper. The experimental results show that, under the appropriate reconstruction temperature, with the increase of reaction time, gas flow rate and magnetic field intensity, the iron recovery will increase and the iron grade will decrease. The most suitable magnetic field intensity is 75 mT, the magnetic material yield is 46.00%, the iron grade is 29.10%, and the iron recovery is 64.12%. Compared with the initial steel slag, the iron grade increased by 8.22%, and the iron recovery increased by 46.38% compared with the direct magnetic separation without oxidation.

**Keywords:** BOF slag; oxidative reconstruction; magnetic separation; iron recovery

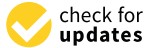



## 1. Introduction

After decades of rapid development, China's iron and steel industry has achieved fruitful results in all aspects. However, due to the inevitable problem of solid waste in heavy industry, iron and steel enterprises have been troubled. According to the statistics of the National Development and Reform Commission, by the end of 2020, China's crude steel output has reached 1.054 billion tons, the production rate of steel-making slag is 8–15% of crude steel, and the emissions of steel-making slag in 2020 are about 84–158 million tons; among them, BOF slag accounts for the vast majority. Due to the large output, large composition fluctuation, poor stability and other reasons, BOF slag cannot be used as cement building materials such as blast furnace slag, so its comprehensive utilization rate is low [1,2].

Stacked BOF slag not only occupies limited land resources, but also pollutes water and soil, which is also a huge waste of resources. In order to make more rational use of BOF slag, it is a key problem that metallurgical enterprises need to find a way to deal with BOF slag on a large scale and recycle it. At present, there are two common methods for recovering iron and iron oxides from BOF slag: reduction and oxidation. The reduction method is mainly to reduce the iron-containing oxides in the BOF slag to metallic iron, and then separate them by magnetic separation. Although metallic iron can be obtained, the reduction process not only consumes a significant amount of energy, but also generates greenhouse gases, which is bad for the long-term development of the environment, and it is also contrary to the goals of carbon peaking and carbon neutrality [3–5].

In recent years, the process of oxidizing steel slag has become the focus of attention. This method oxidizes the non-magnetic phase FeO inside the steel slag into a ferromagnetic

phase $Fe_3O_4$ or $MgFe_2O_4$, and then recovers the iron-containing phase by magnetic separation. Semykina et al. [6,7] respectively analyzed the oxidation mechanism of $Fe^{2+}$ in the $FeO-CaO-SiO_2$ system and the $FeO-CaO-SiO_2-MnO$ system, and found that the required oxygen partial pressure conditions are relatively harsh and cannot be used industrially. Li et al. [8] used steam as an oxidant to oxidize and roast steel slag. After analysis, it was found that it can be transformed into strong magnetic $MgFe_2O_4$, but the method is complicated and the hydrogen generated affects the safety of the experiment. Xue et al. [9] used dry magnetic separation to separate the magnesium-iron spinel phase in the BOF slag after modification, and analyzed the formation mechanism of $MgFe_2O_4$. The addition of $SiO_2$ in the modification reduced the oxidizing atmosphere requirements, but also increased costs. Based on the research group's previous research on BOF slag oxidation [10], this paper explores the optimal magnetic separation process for the strongly magnetic $MgFe_2O_4$ phase in oxidized BOF slag, and solves the problem of iron resource recovery in steel slag without increasing additional cost. In addition, since the main component of the residue after the BOF slag oxidation magnetic separation is $C_2S$ (dicalcium silicate) and other silica-containing calcium phases, it can be used as a raw material for cement and other cementitious materials after a little treatment, and it also contains a large amount of free CaO, MgO and other alkaline oxides. These alkaline substances can also be used as low-cost flue gas desulfurization and denitrification agents. Therefore, the process of steel slag oxidation and magnetic separation can provide a new way for the utilization of BOF slag solid waste, and finally realize a large-scale process application of BOF slag, so as to solve the problem of its low utilization rate [11–15].

## 2. Materials and Methods

### 2.1. Experimental Materials

The experimental slag was obtained from a steel and iron group; samples with a particle size of 140 μm were collected after jaw crushing and electromagnetic crushing, and samples larger than 140 μm were broken several times until all samples were 140 μm and used as BOF slag for the experiment. The composition was analyzed by X-ray fluorescence spectrometer (XRF, PANalytical B.V., Almelo, The Netherlands); its chemical composition is shown in Table 1.

**Table 1.** Chemical analysis of the tested slag.

| Composition | CaO | $SiO_2$ | MgO | $Al_2O_3$ | MnO | $P_2O_5$ | $TiO_2$ | FeO | TFe |
|---|---|---|---|---|---|---|---|---|---|
| Content (wt %) | 41.40 | 16.80 | 6.53 | 4.87 | 3.76 | 1.91 | 1.43 | 20.60 | 20.88 |

### 2.2. Methods

Figure 1 is the phase equilibrium diagram of the oxidized slag calculated by FactSage 7.0 (Thermfact and GTT-Technologies, Montreal and Aachen, Canada and Germany) after the initial steel slag composition in Table 1 is homogenized. The Equilib module was selected in the calculation process, and the product database was selected as Ftoxid-SLAGA, Ftoxid-SPINA, Ftoxid-MeO-A, Ftoxid-b$C_2S$ and Ftoxid-a$C_2S$. The ambient atmospheric pressure was set as a standard atmospheric pressure, the partial pressure of oxygen was 0.21 atm, and the reaction temperature was 800–1500 °C. It can be seen that the spinel phase will appear at a temperature between 1050 and 1400 °C, which means that the temperature range of spinel phase formation is wide, and the FeO phase in steel slag can be transformed into a magnetic spinel phase at a lower temperature.

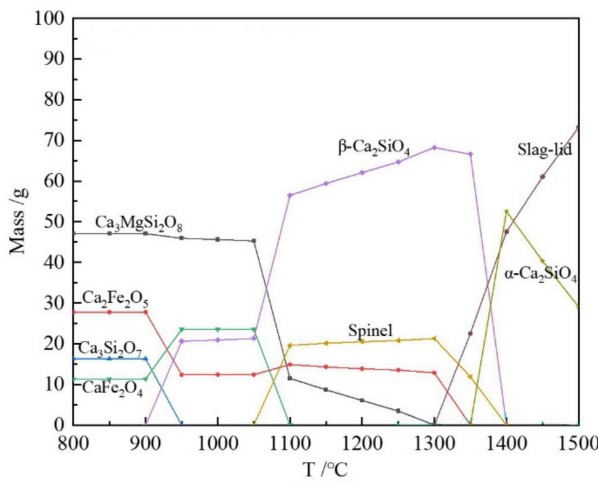

**Figure 1.** Phase equilibrium diagram of slag oxidized.

In this experiment, a sample weighing 50 g was placed in a corundum crucible and was heated by a vertical $MoSi_2$ high-temperature tube furnace. The experimental device was the same as that used in our previous work [10]. During the heating process, argon gas was introduced from the top of the furnace as the protective gas. We first explored the influence of different thermal insulation temperatures (950, 1000, 1050, and 1100 °C) on the experimental results. At this time, the gas flow was controlled at 1 L/min and the thermal insulation time was controlled at 60 min and the injection compressed air as the oxidizing gas. Similarly, we also explored the influence of different flow rates (0.5, 0.75, 1, and 1.25 L/min) on the experimental results. At this time, the insulation temperature was controlled at 1050 °C and the insulation time was controlled at 60 min, the oxidation gas was still selected as compressed air. Finally, we explored the influence of the heat preservation time (20, 40, 60, and 100 min) on the experimental results. At this time, the holding temperature was controlled at 1050 °C, the gas flow was controlled at 1 L/min, and other experimental conditions were the same as above. When the oxidation process was over, the sample was taken out and quickly cooled to room temperature using nitrogen blowing. The experimental conditions of oxidation are shown in Table 2.

**Table 2.** Experimental conditions of oxidation and magnetic separation.

| Procedure | | Temp. (°C) | Time (min) | Flow Rate (L/min) | Magnetic Field (mT) |
|---|---|---|---|---|---|
| Temp. (°C) | 950 1000 1050 1100 | - | 60 | 1 | 50, 75, 100 |
| Time (min) | 20 40 60 100 | 1050 | - | 1 | 50, 75, 100 |
| Flow rate (L/min) | 0.5 0.75 1 1.25 | 1050 | 60 | - | 50, 75, 100 |

The oxidized slag was fully ground to 74 μm by a planetary ball mill. Five grams of the sample was used for magnetic separation, using DTCXG-ZN50 wet magnetic separator (Dongtang Electric Co., Ltd., Tangshan, China). In order to prevent the phase change of steel slag after contact with water, alcohol was selected as the solvent of steel slag. The magnetic

field intensities of magnetic separation are set in Table 2, which are 50, 75 and 100 mT, respectively. After separation, the magnetic material (magnetic separation concentrate) and non-magnetic material (magnetic separation tailings) were collected. The total iron content of magnetic separation concentrate was analyzed, and the recorded quality data, the yield, iron grade and iron recovery rate of the magnetic separation concentrate were calculated, respectively.

Among them, the iron content of the magnetic separation concentrate was determined by the GB/T6730.65-2009 titanium trichloride reduction potassium dichromate titration method. The method of decomposing the sample was the hydrochloric acid-sodium fluoride decomposition method. The sample was decomposed by most of the trivalent iron in the stannous chloride reduction test solution, and then sodium tungstate was used as an indicator, titanium trichloride would reduce all the remaining trivalent iron to divalent to produce "tungsten blue", with dichromic acid potassium solution oxidizing the excess reducing agent. In a sulfuric acid-phosphoric acid medium, using sodium diphenylamine sulfonate as an indicator, potassium dichromate standard titration solution was used to titrate ferrous iron. The expression of the mass fraction of total iron in the magnetic separation concentrate is:

$$\text{Iron grade (\%)} = \frac{c \times (V - V_0) \times 55.85}{m \times 1000} \times 100. \tag{1}$$

In Formula (1): $c$ is the concentration of potassium dichromate standard titrant (mol/L), $V$ is the volume of potassium dichromate standard titrant consumed by the titration sample solution (mL), $V_0$ is the titration blank test solution consumption of potassium dichromate Standard titrant volume (mL), and $m$ is the mass of the sample (g).

The yield expression of magnetic separation concentrate is:

$$\text{Yield (\%)} = \frac{M_1}{M_2} \times 100\%. \tag{2}$$

The expression of iron recovery rate is:

$$\text{Iron recovery rate (\%)} = \frac{TFe_1 \times M_1}{TFe_2 \times M_2} \times 100\%. \tag{3}$$

In Formulas (2) and (3), $TFe_1$ is the mass of total iron in the magnetic material separated by magnetic separation, $M_1$ is the mass of the magnetic material separated by magnetic separation, and $TFe_2$ is always the total mass of the magnetic material. Iron mass, $M_2$ is always the total material mass.

## 3. Results and Discussion

### 3.1. Phase Composition of Raw Slag, Oxide Slag and Magnetic Separation Slag

Figure 2 shows the XRD patterns of the raw slag used in the experiment and the slag under different oxidation conditions. It can be found that the mineral phase system in the raw steel slag is more complicated, including β-$Ca_2SiO_4$ (β-$C_2S$) and $Ca_3SiO_5$ ($C_3S$), $Ca_3MgSi_2O_8$ ($C_3MS_2$), FeO, $Fe_3O_4$ and a small amount of $Ca_2Fe_2O_5$ ($C_2F$). After oxidation under certain conditions, the number of phases is reduced, mainly composed of silicon-calcium phase and magnesium-iron spinel phase. It can also be seen from the figure that when the oxidation time is increased and the air flow rate is increased, the diffraction peak intensity of $MgFe_2O_4$ increases, indicating that, as the degree of gas-solid reaction deepens, the content of the $MgFe_2O_4$ spinel phase increases. This is mainly due to the diffusion of gas molecules from the upper sample to the lower sample during the oxidation process until the sample is completely oxidized.

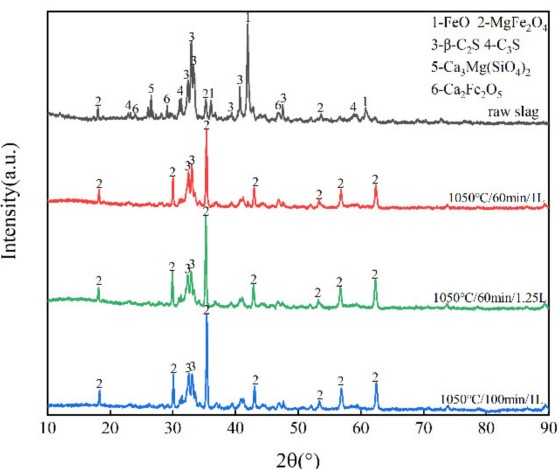

**Figure 2.** X-ray pattern of raw and oxidized slag under different conditions.

In the oxidation process of raw slag, FeO in slag is oxidized into $Fe_3O_4$ and $Fe_2O_3$; the $Fe_2O_3$ produced will form a solid solution $MgFe_2O_4$ with the MgO in the slag. Since the diameters of the two ions of $Mg^{2+}$ and $Fe^{2+}$ are very close, and the radius of $Mg^{2+}$ is slightly larger than that of $Fe^{2+}$, in a certain temperature range, $Mg^{2+}$ will enter the $Fe_3O_4$ lattice through solid-state diffusion to replace part of $Fe^{2+}$, and then $MgFe_2O_4$ is formed and a more stable spinel phase is precipitated. The above two methods are the main methods for oxidizing BOF slag to form strong magnetic $MgFe_2O_4$ [16–20].

The XRD patterns of the magnetic separation concentrate and magnetic separation tailings after magnetic separation are shown in Figure 3. It can be found that the magnetic separation concentrate phase is mainly composed of $MgFe_2O_4$ and $\beta$-$C_2S$. Compared with the diffraction peaks corresponding to $MgFe_2O_4$ and $\beta$-$C_2S$ in the oxide slag XRD, the $MgFe_2O_4$ diffraction peak intensity is higher, while the $\beta$-$C_2S$ diffraction peak intensity is lower, which means that the $MgFe_2O_4$ phase is the main phase in the magnetic separation concentrate. Therefore, the magnetic separation concentrate can be used for ironmaking or sintering. The phases in the magnetic separation tailings are composed of $MgFe_2O_4$, $\beta$-$C_2S$ and FeO. Compared with the magnetic separation concentrate, the diffraction peak of $MgFe_2O_4$ has a much lower intensity, which also shows that the strong magnetic $MgFe_2O_4$ can be separated by magnetic separation.

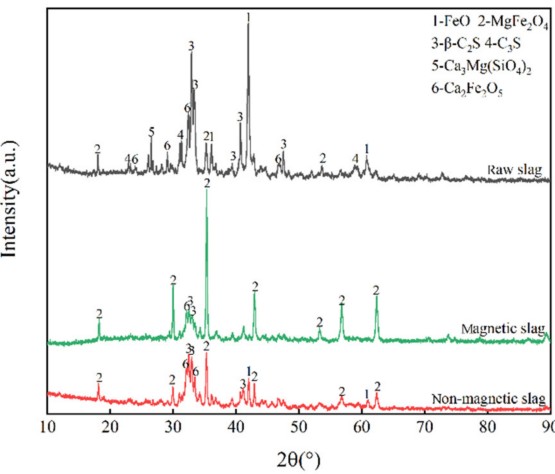

**Figure 3.** X-ray pattern of raw and magnetic separation slag.

### 3.2. The Influence of Oxidation Temperature on the Effect of Magnetic Separation

Firstly, the influence of temperature on the magnetic separation effect was investigated. When 1 L/min of air is introduced, the raw slag is oxidized at different temperatures for 60 min, then the sample is taken out and ground to 200 mesh, and magnetic separation is carried out at a magnetic field strength of 75 mT. The yield, iron grade and iron recovery of the magnetic separation concentrate of the 950–1100 °C oxide slag are shown in Figure 4. It can be found that when the oxidation temperature is greater than 1000 °C, the iron grade fluctuates between 27.32% and 27.74%, and the iron recovery fluctuates between 75.53% and 76.00%. The overall magnetic separation effect is not much different, which shows that, within the appropriate reaction temperature range, the amount of $MgFe_2O_4$ produced is not much different, and it also proves from the side that the choice of the reaction temperature range is correct.

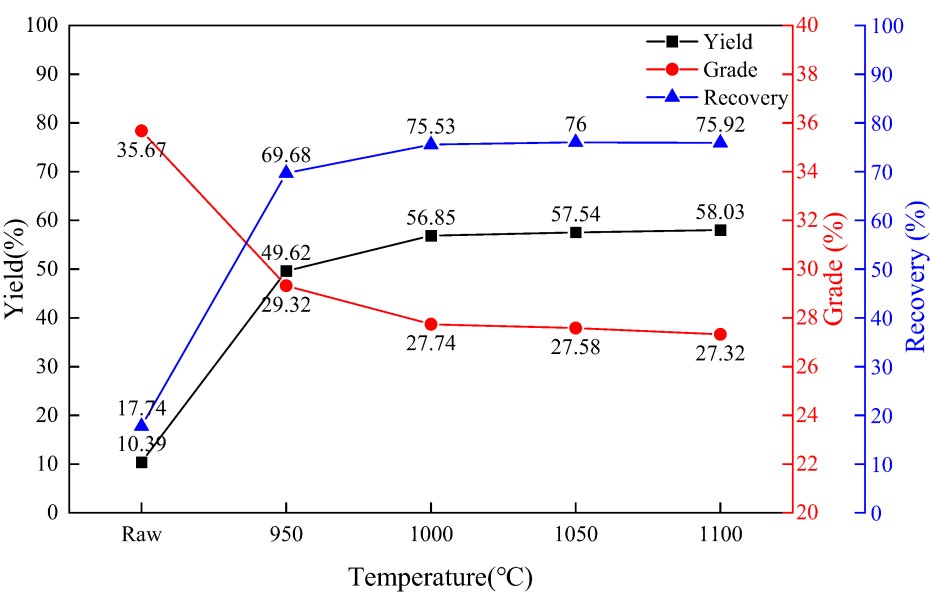

**Figure 4.** Magnetic separation effect between different oxidation temperatures at 75 mT.

### 3.3. The Influence of Gas Flow Rate on the Effect of Magnetic Separation

It can be seen from Figure 4 that the magnetic separation effect of BOF slag is the best after 60 min of oxidation at 1050 °C. Therefore, the influence of different gas flow rates (0.5, 0.75, 1, and 1.25 L/min) on the magnetic separation effect is investigated under this temperature condition. The magnetic separation effect is shown in Figure 5. The figure shows that, with the increase of the flow rate, the magnetic separation yield and iron recovery are increasing, which is far better than the magnetic separation result of the raw slag, indicating that the appropriate amount of air can oxidize the slag to a greater extent. Among them, the yield of magnetic separation concentrate is the highest at 1.25 L/min. However, due to the sharp drop in grade, the recovery is reduced. When the gas flow rate is 1 L/min, the recovery is the highest, and the iron grade is relatively high, which is a suitable gas flow rate. When the gas flow rate increases, the oxidation reaction speed is increased, and the conversion of FeO inside the sample to $MgFe_2O_4$ is intensified. The amount of formation is the largest, but the magnetic separation effect decreases. This is because with the progress of the reaction, the $Fe^{2+}$ inside the sample is gradually oxidized to $MgFe_2O_4$, and the magnetic separation concentrate selected during the magnetic separation process increases, resulting in an increase in the yield and recovery, and $MgFe_2O_4$ is wrapped by $C_2S$. This can also be demonstrated by a scanning electron microscope (SEM, Carl Zeiss AG, Oberkochen, Germany) photograph of oxidized BOF slag in Figure 6. The non-magnetic phases are selected together by the magnetic field, resulting in a decline in grade.

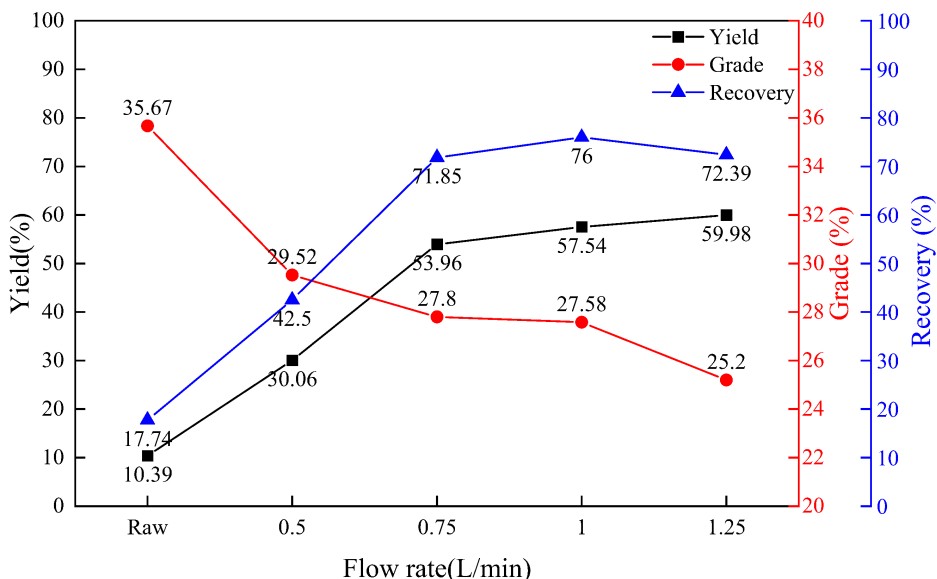

**Figure 5.** Magnetic separation effect between different gas inlet flow rates at 75 mT.

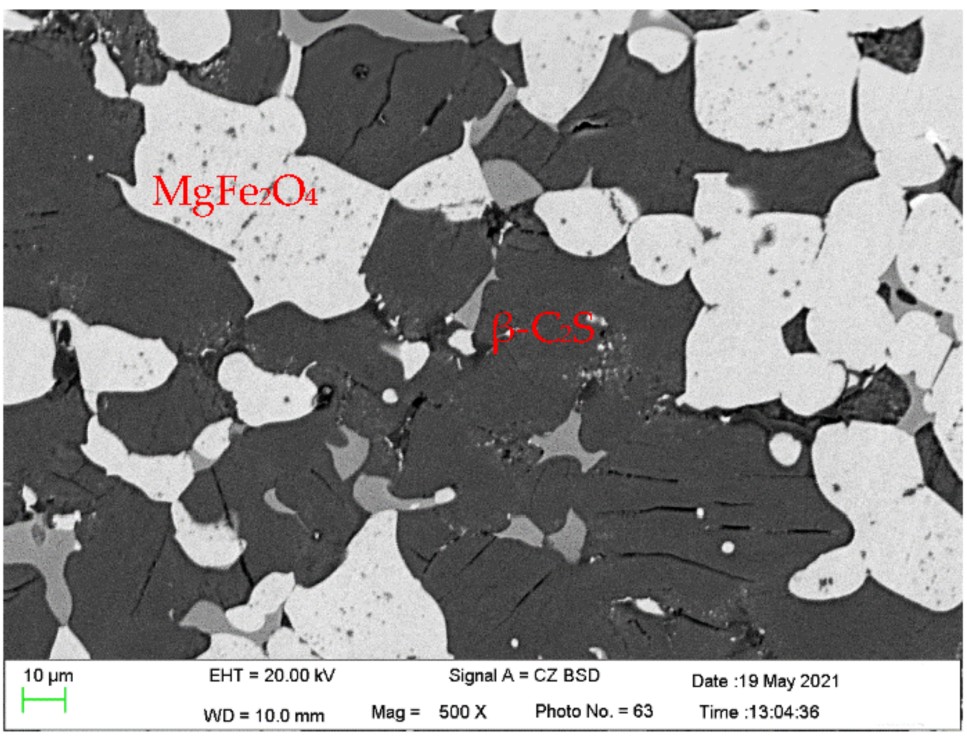

**Figure 6.** Scanning electron microscope photograph of oxidized BOF slag.

### 3.4. The Influence of Oxidation Time on the Effect of Magnetic Separation

It can be seen from Figures 4 and 5 that the magnetic separation effect is best when the oxidation temperature is 1050 °C and the gas flow rate is 1 L/min. Therefore, the comparison of the magnetic separation effect of oxidized BOF slag when holding for 20–100 min is shown in Figure 7. The effect of magnetic separation is similar to that of different flow rates. The relationship between yield and iron recovery and reaction time is positively correlated, while grade is negatively correlated. This shows that the nature of time and flow changes are the manifestation of the reaction process, which corresponds

exactly to the XRD pattern of the steel oxide slag. Finally, as the reaction deepens, the magnetic separation effect first increases and then decreases.

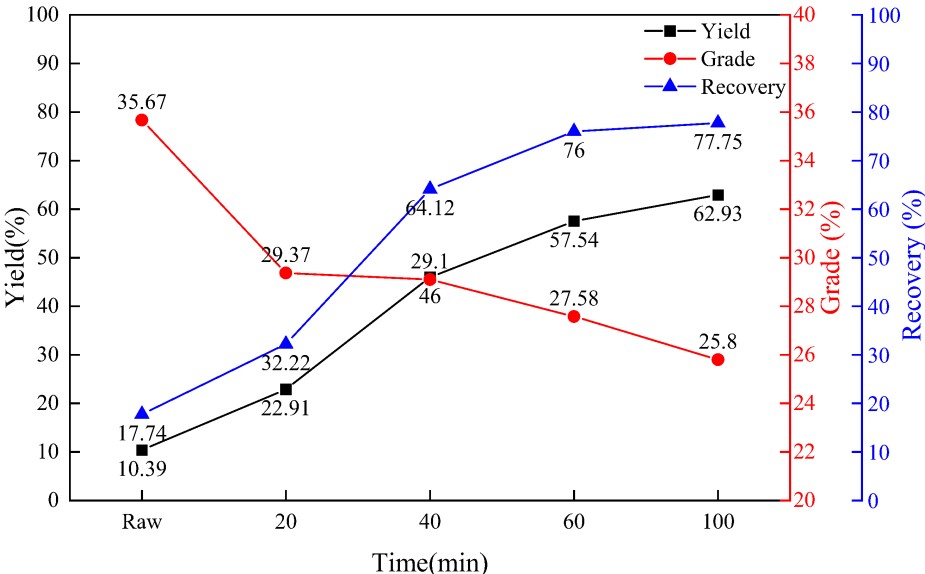

**Figure 7.** Magnetic separation effect under different oxidation time at 75 mT.

### 3.5. The Influence of Magnetic Field Strength on the Effect of Magnetic Separation

Finally, the influence of magnetic field on magnetic separation effect was explored. The optimal oxidation conditions were selected, that is the gas flow rate was 1 L/min, and the steel slag was oxidized for 60 min at 1050 °C, the oxide slag was taken out, and wet magnetic separation was used to perform magnetic separation under different magnetic fields (50, 75, 100 mT). Under different magnetic fields, the yield, iron grade and iron recovery of the magnetic separation concentrate are shown in Figure 8. It can be seen from the figure that as the magnetic field strength increases, the yield and iron recovery increase, while the iron grade is falling. $MgFe_2O_4$ is a strong magnetic phase, which can be selected in a weaker magnetic field. From the microstructure of the oxide slag, it is known that the $MgFe_2O_4$ spinel phase is embedded in the base phase $\beta$-$C_2S$, so it will be brought out when $MgFe_2O_4$ is selected. In the process of slag oxidation, some $MgFe_2O_4$ spinel phase particles are smaller and are covered by more $\beta$-$C_2S$, and cannot be selected under a smaller magnetic field. When the magnetic field is increased, this part is selected, the iron recovery was improved, and the increase of $\beta$-$C_2S$ also led to the decrease of iron grade. When the magnetic field is increased to 100mT, almost all the magnetic phase is recovered, and the iron recovery is almost 100%. In actual production, the relationship between the yield, grade and iron recovery of the magnetic separation concentrate should be considered. That is to say, a higher iron recovery can be obtained in the case of a lower yield, and the grade is in a moderate range, so 75 mT is a more suitable magnetic separation intensity.

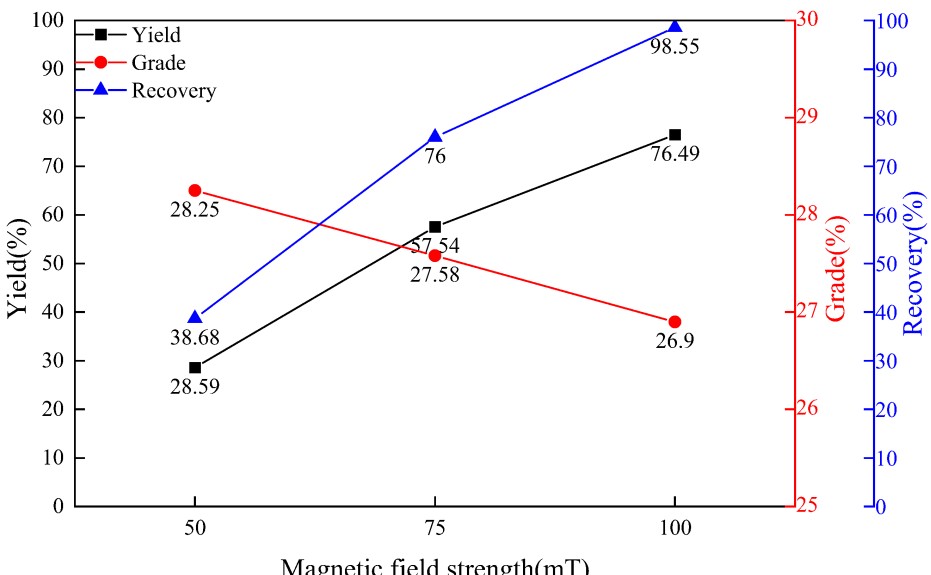

**Figure 8.** Magnetic separation effect under different magnetic field conditions.

## 4. Conclusions

(1) Through oxidation treatment in an air atmosphere, and under suitable conditions of reaction temperature, time and air flow rate, the magnetic iron oxides in the steel slag can be transformed into ferromagnetic magnesium-iron spinel;

(2) When the reaction temperature is controlled between 1050 and 1100 °C, the oxidation time is controlled between 40 and 60 min, the air flow rate is controlled at 0.75–1 L/min and the magnetic field strength is 75 mT, the yield of modified BOF slag is 46–57.54%, the iron grade can reach 27.58–29.10%, and the iron recovery can reach 64.12–76.00%, which is the best process parameter range for the magnetic separation experiment;

(3) The magnetic field strength has a great influence on the results of magnetic separation. The effect is best when the magnetic separation strength is 75 mT. After 75 mT, the magnetic separation yield will rise sharply, the grade of the concentrate will decrease, and the magnetic separation effect does not change significantly with the increase of the magnetic field strength.

**Author Contributions:** M.L. and Z.H. prepared and revised the manuscript under the supervision of X.H. All authors contributed to the general discussion. All authors have read and agreed to the published version of the manuscript.

**Funding:** This work is financially supported by State Key Laboratory of Advanced Metallurgy, University of Science and Technology Beijing (Grant No. 41620024 and 41622012).

**Informed Consent Statement:** Not applicable.

**Data Availability Statement:** Not applicable.

**Conflicts of Interest:** The authors declare no conflict of interest.

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
