# Peer review of "Optimization of Iron Recovery from BOF Slag by Oxidation and Magnetic Separation"

_metals, doi:10.3390/met12050742_

Round 1

Reviewer 1 Report

The article deals with the influence of oxidation temperature, oxidation time, flow rate of compressed air  and magnetic induction on the yield  (probably) of concentrate with magnetic properties, iron recovery and iron recovery grade.

The value of the article is very often degraded by incorrect or incomprehensible terms.

My questions and comments on the article:

  • In the article, the authors state: "The experimental slag was obtained from a steel and iron group" but in the title of the article is mentioned BOF slag.
  • Table 2 is a bit incomprehensible because the number of experiments is not entirely clear. 
  • I suggest editing the table to make the experimental parameters clear.
  • How many experiments were performed?
  • "Weigh 5 g sample, the magnetic separation equipment is DTCXG-ZN50 wet magnetic separator" I don´t understand this sentence.
  • Analyzed the chemical composition of the magnetic separation concentrate, and combined the recorded quality data, the yield, iron content and iron recovery rate of the magnetic separation concentrate were calculated, respectively" I don't quite understand the given sentence and I can't find the chemical analysis of the concentrate anywhere
  • What is "Magnesia feldspar (C2MS2)"  and "Fustenite (FeO)"? C2MS2  is akermanite and FeO is wűstite.
  • According to Fig. 3, there is only the MgFe2O4 phase.
  • Incomprehensible descriptions of Y axes (Fig. 4, 5, 6 and 7).
  • I recommend change the  the figures names.
  • Use full term in text and in figures :Yield (of what), grade (of what), recovery (of what).
  • What is "the yield of modified steel slag is 46 ~ 240
    57.54%, "

Reviewer 2 Report

The paper is interesting and presents important information on how to treat BOF waste to maximise the utilisation and reuse of this waste material. There are a few points that I think should be clarified in the paper to help with understanding, particularly within the materials and methods section.

Introduction:

  • Please define ‘BOF’
  • Line 41: replace ‘a lot of energy’ with ‘a significant amount of energy’
  • Line 41: replace ‘a lot of greenhouse gases’ with ‘greenhouse gases’
  • Line 44 remove ‘everyone’s’
  • Line 51: replace ‘device’ with ‘method’
  • Line 67 – remove ‘completely’

Materials and methods.

2.1 Experimental methods

  • Line 70 - Is it possible to give details about the source of the BOF? As you state in the introduction this material is highly variable.
  • Line 71 -when using a sieve, typically it is stated that the particle size is ‘less than 140 µm’ for example. Better to present particle size as µm (instead of mesh of sieve). Please rephrase ‘particle size of 100 mesh’ to ‘particle size less than xxx µm’ (I believe this is 140 µm for 100 mesh) throughout the text.
  • Line 73 - how was the chemical composition was measured?

2.2 Methods

  • Line 77: Please can you provide further details on the Factsage calculation. What database was used? What were the condition for the calculation set up? Is this a calculation (Figure 1) for the fixed composition (in Table 2) with changing temperature, or have you introduced oxygen into the calculation? This was not clear in the description.
  • Line 87: heating rate?
  • Line 88: ‘heat preservation’ – do you mean ‘isothermal hold’
  • Line 95 – what were the conditions for the planetary ball milling (container, media, speed time?)
  • Line 95-96 rephase ‘5g of the sample was used for magnetic separation, using DTCXG-ZN50 wet magnetic separator.’
  • Line 101-103 – please reword to make clearer.

Results and discussion

3.1

  • how did the phases identified using XRD correspond to the Factsage database calculation?
  • Line 134 - Reword: “After oxidation under certain conditions, the phase composition of the oxide slag is reduced”, to “After oxidation under certain conditions, the number of phases is reduced”
  • Line 136 - “It can also be seen from the figure that when the oxidation time is increased and the air flow rate is increased, the diffraction peak intensity of MgFe2O4 increases, indicating that as the degree of gas-solid reaction deepens, the content of the MgFe2O4 spinel phase increases”

I have to say that this is very difficult to observe in the Figure 2. The 100% peak for spinel phase appears to me to be similar in all samples. Is this difference significant?

3.3 The influence of gas flow rate on the effect of magnetic separation

  • Line 192: “This is because that with the deepening of the reaction, the Fe2+ inside the sample is gradually oxidized to MgFe2O4, and the magnetic separation concentrate selected during the magnetic separation process increases, resulting in an increase in the yield and recovery, and MgFe2O4 is wrapped by C2 The non-magnetic phases are selected together by the magnetic field, resulting in a decline in grade.”

Can you please explain this part further, as it is the first time it is mentioned that the C2S coats/surrounds the MgFe2O4. What evidence is there for this? Is there microscopy data for example showing layer formation?

  • Line 192: Replace “deepening of the reaction” with ‘progress of the reaction’?

Reviewer 3 Report

A rather interesting study is presented that is devoted to a problem that has not been solved to date - the problem of deep processing of steel-smelting slags with the extraction of most of the oxidized metal from the slag. The results are obtained, which certainly can be of considerable interest to specialists. I can recommend publishing this paper.
However, I have a number of comments that would be useful to take into account in order to improve the paper.
1.  I would recommend supplementing the literary review part of the work with a paper 10.3844/ajassp.2015.952.961 that is quite close to the topic of this study.  
2. What is "TFe" in Table 1?
3. The authors have the opportunity to implement thermodynamic modeling in the FactSage 7.0 program. Why didn't they run simulations for oxidizing conditions (involving the oxygen-containing gas phase)? This would be more illustrative than only modeling solid phases (Figure 1).
4. The process of thermodynamic calculation of Figure 1 should have been described in more detail (what bases were used?).
